# Sixteen Years of HPV Vaccination in Mexico: Report of the Coverage, Procurement, and Program Performance (2008–2023)

**DOI:** 10.3390/ijerph22071028

**Published:** 2025-06-27

**Authors:** Rodrigo Romero-Feregrino, Raúl Romero-Cabello, Raúl Romero-Feregrino, Paulina Vilchis-Mora, Berenice Muñoz-Cordero, Mario Alfredo Rodríguez-León

**Affiliations:** 1Asociación Mexicana de Vacunología, Instituto Para el Desarrollo Integral de la Salud (IDISA), Instituto Mexicano del Seguro Social (IMSS), CONCAMIN, Saint Luke School of Medicine, Academia Mexicana de Pediatria, Av. Cuauhtémoc 271, Interior 101, Colonia Roma, Cuauhtémoc, Mexico City ZC 06700, Mexico; drrodrigo@idisalud.com; 2Asociación Mexicana de Vacunología, Instituto Para el Desarrollo Integral de la Salud (IDISA), Hospital General de México, Department of Microbiology and Parasitology, Universidad Nacional Autónoma de México (UNAM), Saint Luke School of Medicine, Academia Mexicana de Pediatria, Av. Cuauhtémoc 271, Interior 101, Colonia Roma, Cuauhtémoc, Mexico City ZC 06700, Mexico; 3Asociación Mexicana de Vacunología, Instituto Para el Desarrollo Integral de la Salud (IDISA), UMAE Hospital de Pediatría “Dr. Silvestre Frenk Freund”, Centro Médico Nacional Siglo XXI, Instituto Mexicano del Seguro Social (IMSS), Saint Luke School of Medicine, Academia Mexicana de Pediatria, Av. Cuauhtémoc 271, Interior 101, Colonia Roma, Cuauhtémoc, Mexico City ZC 06700, Mexico; 4Saint Luke School of Medicine, Sierra Mojada 415, Lomas de Chapultepec III Secc, Miguel Hidalgo, Mexico City ZC 11000, Mexico; 5Asociación Mexicana de Vacunología, Instituto Para el Desarrollo Integral de la Salud (IDISA), Pediatrics Department/Hospital General de Cuajimalpa IMSS-Bienestar, Av. Cuauhtémoc 271, Interior 101, Colonia Roma, Cuauhtémoc, Mexico City ZC 06700, Mexico; bere.mu83@gmail.com; 6Asociación Mexicana de Vacunología, Instituto Para el Desarrollo Integral de la Salud (IDISA), Universidad Nacional Autónoma de México (UNAM), Av. Cuauhtémoc 271, Interior 101, Colonia Roma, Cuauhtémoc, Mexico City ZC 06700, Mexico

**Keywords:** vaccination, human papilloma virus, coverage, Mexico, public health, human papillomavirus (HPV) vaccination, immunization program evaluation, vaccine coverage assessment, HPV vaccine, public health surveillance, cervical cancer prevention strategies, health information systems and data quality, vaccination program performance in Mexico, papillomavirus vaccines, immunization programs, vaccination coverage, health services accessibility, public health surveillance, uterine cervical neoplasms/prevention and control, health information systems, Mexico/epidemiology

## Abstract

Introduction: In 2008, Mexico initiated its national HPV vaccination program targeting adolescent girls. This study aims to evaluate the current status of the program, analyzing trends in vaccine acquisition, administration, and coverage over a 16-year period. Materials and Methods: A retrospective longitudinal study was conducted using secondary data from 2008 to 2023. Official records from three major public health institutions—IMSS, ISSSTE, and SSA—were reviewed to assess HPV vaccine procurement and administration. Results: Significant fluctuations were identified in the number of doses acquired, administered, and the corresponding coverage rates. A marked decline was observed between 2019 and 2021, followed by a sharp increase in 2022 and 2023. Over the entire period, an estimated 6.8 million doses were not administered to the intended target population. Furthermore, 2.6 million doses were administered in excess of the number officially acquired, indicating possible discrepancies in data reporting or vaccine inventory management. Discussion: The findings revealed substantial inconsistencies in vaccine procurement, administration, and coverage across institutions. While IMSS and ISSSTE consistently reported coverage below the theoretical target, SSA occasionally exceeded expectations, potentially compensating for deficits elsewhere. Nevertheless, national coverage remained inadequate in several years, with notable disparities between institutions. These gaps highlight systemic weaknesses in program coordination, planning, and data transparency, contributing to millions of unvaccinated individuals. Conclusions: This study offers a comprehensive analysis of Mexico’s HPV vaccination program, uncovering critical irregularities in its implementation. Challenges include inaccurate target population estimation, inconsistencies between vaccine acquisition and administration, and limited data reliability. Despite some progress in recent years, particularly in the post-pandemic years, the program requires urgent restructuring. This includes implementing a national catch-up strategy, expanding vaccine eligibility, and strengthening surveillance systems to ensure equitable and effective coverage toward the elimination of cervical cancer.

## 1. Introduction

The first prophylactic vaccine targeting human papillomavirus (HPV)-associated diseases was authorized for use in 2006 in the USA and Europe [Quadrivalent HPV L1 VLP vaccine (Gardasil-4)]. To date, six HPV vaccines have been licensed globally. These vaccines are designed for administration prior to HPV exposure, ideally before the initiation of sexual activity. They are indicated for the prevention of cervical intraepithelial neoplasia and cervical cancer associated with high-risk HPV genotypes, although the specific HPV types covered vary by vaccine formulation. Furthermore, based on their respective regulatory approvals, certain HPV vaccines also have indications for the prevention of additional HPV-related diseases [1].

Efficacy in women with no evidence of infection prior to vaccination is with bivalent vaccine 93% against CIN grade 2 or 3, and quadrivalent vaccine 87% against adenocarcinoma in situ, 95% against any grade of intracervical neoplasia, and 99% for genital warts [2].

The principal objective of HPV vaccination is the prevention of cervical cancer, which represents approximately 82% of all malignancies attributable to HPV infection. In line with this goal, the World Health Organization’s 2020 Global Strategy to Accelerate the Elimination of Cervical Cancer as a Public Health Problem by the year 2030 established a global target to reduce cervical cancer incidence to fewer than four cases per 100,000 women [1,3].

The HPV vaccine was introduced in Mexico in 2008 for girls from 12 to 16 years of age in 125 municipalities with the lowest human development index; in 2009, the schedule was approved (0–6–60 months); and in 2012, universal vaccination began for girls from 5th grade of primary school and 11 years of age not attending school with the vaccination scheme of 0, 6, and 60 months. In 2014, the vaccination schedule was modified from 3 to 2 doses, eliminating the 60-month dose; in 2021, the vaccination schedule was modified to 1 dose; and in 2023, it was further modified to administer 1 dose to females aged 11, 12, and 13 years [2].

In summary, official recommendations in Mexico indicated the vaccination of 11-year-old girls until 2022 [2]. In 2023, these indications were expanded to include girls aged 10, 11, and 12.

In terms of coverage, in Mexico, the Pan-American Health Organization reports a coverage of 0.5% in women in 2021 and 0% in men, 50% in 2020, 96.83% in 2019, and 98% in 2018 for indications of the national vaccination program [4,5]. The National Health and Nutrition Survey carried out by the Ministry of Health of Mexico reports coverage of 43.7% in 2022 [6].

The new cases reported of human papilloma virus infection (HPVi) in Mexico in Table 1, with mild and moderate cervical dysplasia (CIN1-2), severe cervical dysplasia, and cervical cancer in situ (CIN 3/CIS, cervical cancer (CC)) were as follows [7]:

Previous analyses of the Mexican National Immunization Program—specifically regarding influenza and Bacillus Calmette–Guérin (BCG) vaccines—have identified various inconsistencies within official datasets [8,9]. In light of these reports and our contextual understanding of vaccinology and the Mexican healthcare system, we deemed it necessary to conduct a detailed review of the official records related to vaccine acquisition, administration, and coverage. The objective was to assess the current status of human papilloma virus vaccination efforts at the national level.

This study aims to describe and analyze the available data to provide a clearer understanding of the program’s current performance, and to generate actionable recommendations for addressing identified gaps. Central to this investigation are the following questions: What is the actual vaccination coverage? And what strategies can be implemented to improve it?

We hypothesize that the low concordance between official HPV vaccination figures reported by public health institutions and the estimates derived from independent analyses reflects underlying systemic and multifactorial challenges. These include institutional differences in operational capacity, disruptions caused by the COVID-19 pandemic, inaccurate estimation of the target population, the absence of a national nominal vaccination registry, frequent changes in vaccination guidelines, and inadequate inventory management systems. This variability and inconsistency in data reporting highlight the urgent need for evidence-based strategies to strengthen and standardize the national HPV immunization program in Mexico.

## 2. Material and Methods

Study Design. This was a retrospective, longitudinal, and ecological study based on secondary data obtained from official government sources. The study period covered 16 years, from 2008 through 2023.

Data Sources. Data were collected from the three main public health institutions in Mexico:⚬The Mexican Social Security Institute, Mexico (Instituto Mexicano del Seguro Social, IMSS),⚬The Institute for Social Security and Services for State Workers, Mexico (Instituto de Seguridad y Servicios Sociales de los Trabajadores del Estado, ISSSTE), and⚬The Ministry of Health, Mexico (Secretaría de Salud, SSA).

Together, these institutions provide healthcare services to approximately 98% of the Mexican population. According to 2020 estimates, this includes more than 128 million inhabitants and approximately 2.2 million live births annually [10,11,12,13,14,15].

Variables and Data Collection. The primary variables analyzed included the following:⚬Number of vaccine doses acquired or purchased,⚬Number of doses administered or apply,⚬Vaccine coverage rates.

Data were extracted from publicly available institutional reports, annual health statistics, and official registries. Only validated, government-issued datasets were used to ensure data quality and reliability.

Analytical Approach. Descriptive and comparative analyses were performed to identify trends in vaccine acquisition and administration over time. The data were used to construct theoretical models capable of explaining observed patterns and predicting future performance. These models supported the identification of gaps and inefficiencies within the immunization program.

Ethical Considerations. As this study was based entirely on secondary, publicly available data without individual patient identifiers, it did not require ethical review or informed consent, in accordance with national and international guidelines for research involving de-identified datasets.

Data Requirements and Sources. To carry out the analysis, the following specific data elements were identified as essential: the number of beneficiaries per institution, detailed information on each vaccine (including type and indication), the number of doses procured, and the number of doses administered.

Data on vaccine procurement were obtained through formal requests submitted to the National Institute for Transparency, Access to Information, and Protection of Personal Data (INAI, by its Spanish acronym), and were retrieved from its official databases [16,17,18,19,20,21,22,23,24,25,26,27,28,29]. Information regarding vaccine administration was sourced from the historical data repositories of each institution: IMSS [30], ISSSTE [31], and SSA [32].

Data Processing and Indicators. Following data acquisition, analytical models were developed to enable comparative assessments across institutions and time periods. Several key metrics were defined to guide the data interpretation as follows:⚬Theoretical Target Population: Defined as the estimated number of individuals eligible for vaccination, based on the specific vaccine indications and the population assigned to each healthcare institution.⚬Annual Acquisitions and Year-over-Year Variation: Refers to the annual count of vaccine doses procured and the percentage change relative to the previous year.⚬%PUR (Procurement-to-Target Ratio): The percentage of vaccine doses acquired in relation to the theoretical target population.⚬%APP (Application-to-Procurement Ratio): The percentage of vaccine doses administered relative to the number of doses acquired.⚬%COV (Coverage Rate): The percentage of vaccine doses administered with respect to the theoretical target population.

The estimated target population in Table 2, was derived by integrating demographic projections from the National Population Council (CONAPO) with the criteria established in national immunization guidelines.

For each healthcare institution, the theoretical target population was estimated through a formula that incorporated the total national target population alongside the proportion of the population served by each respective institution (IMSS, ISSSTE, and SSA). It is called theoretical because it is based on CONAPO population estimates. The percentage distribution of the beneficiary population across institutions is detailed in Table 3.

Institution-specific target population estimates were derived using demographic data [10,11,12,13,14,15] and the following formulae:⚬Total theoretical target population: Defined as the number of 11-year-old girls for all years except 2023, in which the eligible population included girls aged 10, 11, and 12 years.⚬IMSS: Total theoretical target population × proportion of the population affiliated with IMSS per year.⚬ISSSTE: Total theoretical target population × proportion of the population affiliated with ISSSTE per year.⚬SSA: Total theoretical target population × proportion of the population covered by SSA per year.

To analyze the trends in vaccine procurement, the annual percentage change in the number of doses acquired by each institution was calculated using the following formula:[(Doses acquired in year X − doses acquired in the previous year)/doses acquired in the previous year] × 100

Three indicators were used to evaluate institutional performance in procurement and delivery as follows:⚬%PUR (Procurement-to-Target Ratio): (Number of doses acquired/theoretical target population) × 100⚬%APP (Application-to-Procurement Ratio): (Number of doses administered/number of doses acquired) × 100⚬%COV (Coverage Rate): (Number of doses administered/theoretical target population) × 100

This study analyzed data by calendar year (January–December), and calculated the number of vaccinated and unvaccinated girls in each national cohort each year.

All data processing and statistical calculations were conducted using Microsoft Excel. Figures and visualizations were generated by combining selected variables; only the most representative data are presented.

Additionally, the number of unvaccinated individuals was estimated per institution, and in total, by subtracting the number of administered doses from the theoretical target population. The resulting coverage percentages (%COV) were compared with benchmarks established by the World Health Organization (WHO) [33].

This study presents several methodological limitations. As a retrospective analysis, the research relies on the availability and accuracy of historical institutional records, which may contain inconsistencies, omissions, or reporting delays beyond the control of the researchers. The ecological nature of this study, in which data are aggregated at the institutional level rather than at the individual level, limits the ability to draw causal inferences. Additionally, the use of secondary data restricts the scope of analysis to variables that were originally collected for administrative rather than research purposes, potentially compromising the completeness and precision of the information.

## 3. Results

Data on vaccine procurement were compiled for each institution across a maximum of 16 years, with each data point representing a single year. Thus, full data availability for an institution would correspond to 16 points, or 100% completeness. As shown in Figure 1, data availability varied by institution: the Ministry of Health (SSA) reported data for 9 years (56%), ISSSTE for 11 years (69%), and IMSS for 10 years (63%).

Overall, more than 50% of the expected data were successfully retrieved for each institution, with the exception of the IMSS during the periods 2008–2012 and 2021; ISSSTE in the years 2009, 2014, 2015, 2019, and 2021; and SSA in 2008–2010, 2012, 2018, and 2021. For certain years, it was not possible to determine whether the absence of data was due to a lack of procurement or the non-existence of records.

Figure 1 illustrates the annual number of HPV vaccine doses acquired by each institution. A comparative analysis was conducted between annual acquisitions per institution and the aggregated national total. Notable interannual variability was observed in the number of doses procured, reflecting fluctuations in institutional procurement behavior and programmatic priorities.

Data on annual vaccine administration were available for the majority of the study period. Specifically, the Ministry of Health (SSA) reported 14 out of 16 possible data points (88%), ISSSTE reported 15 points (93%), and IMSS reported 12 points (75%). Most data were successfully recovered, with exceptions noted in 2008 across all three institutions, 2009 for SSA, and the period 2009–2010 for IMSS (Figure 2).

As shown in Figure 2, the number of HPV vaccine doses administered exhibited substantial year-to-year variability both within and across institutions, as well as in the aggregated national total.

Figure 3 presents the calculated percentages for the procurement-to-target ratio (%PUR), the application-to-procurement ratio (%APP), and the coverage rate (%COV) for each institution over the study period.

For IMSS, between 2012 and 2023, %PUR remained below 80% in most years, with the exception of 2014 and 2022, when it reached 100%. The %APP did not exceed 90% in any year except 2018, when it reached 127%. The %COV consistently remained at or below 70% throughout the period.

In the case of ISSSTE, from 2008 to 2023, %PUR was consistently below 50%. The %APP showed marked variability, ranging from 31% to 127%, with six years registering values above 100%, indicating that more doses were administered than officially acquired. The %COV remained below 60% across all years.

For SSA, %PUR was notably high between 2013 and 2017, exceeding 150%, and peaking at 268% in 2023. By contrast, during the period from 2018 to 2022, %PUR fell sharply, ranging from 0% to 30%. The %APP fluctuated widely, from 0% to 1275%, with four years exceeding 100%. Similarly, %COV varied between 3% and 276% from 2006 to 2023, surpassing 100% in eight of those years.

When aggregating data from all three institutions, between 2012 and 2023, %PUR ranged from 0% to 124%, with five years showing values above 100%. The %APP varied from 0% to 1909%, with eight years exceeding 100%. The %COV across the combined institutions ranged from 4% to 113%.

Figure 4 illustrates a comparative analysis between the number of HPV vaccine doses procured (%PUR), the theoretical target population (Obj), and the number of doses administered across the three main institutions.

For IMSS and ISSSTE, both the number of doses acquired and those administered consistently remained below the estimated target population throughout the study period. By contrast, for SSA, between 2014 and 2023, the number of doses administered exceeded the theoretical target population in most years, with the exception of 2020 and 2021, during which administration fell below target levels.

When analyzing all institutions collectively, certain years demonstrated a shortfall in both procurement and administration relative to the target population, while in other years, the number of doses administered surpassed even the number of doses acquired, suggesting possible discrepancies in reporting or the use of previously stocked vaccines.

Figure 5 presents the estimated vaccine coverage (%COV), calculated as the proportion of administered doses relative to the theoretical target population, disaggregated by institution and shown in aggregate.

For IMSS and ISSSTE, coverage levels remained consistently low across most years of the study period. By contrast, SSA reported coverage exceeding 100% in several years, with the exception of 2012, 2013, 2020, and 2021, during which coverage fell below expected levels.

When considering all institutions combined, adequate coverage—defined as meeting or exceeding the theoretical target population—was achieved in approximately half of the years analyzed. In some years, coverage surpassed 100%, while in the remaining years, coverage fell short of the target, reflecting variability in program performance and possible inconsistencies in procurement and delivery.

Figure 6 illustrates that the unvaccinated population was present across all three institutions throughout the study period. For the combined data from IMSS, ISSSTE, and SSA between 2012 and 2023, it was estimated that 6,852,030 vaccine doses were not administered. Of the 25,834,410 doses required to achieve full coverage of the theoretical target population, only 19,219,848 doses were administered. Of these, 17,902,065 doses were confirmed as procured according to the available data.

IMSS and ISSSTE reported the unvaccinated population in every year analyzed. By contrast, SSA showed periods in which the number of vaccines administered exceeded the theoretical target population—specifically from 2014 to 2019 and again in 2022 and 2023—indicating apparent over coverage.

The number of vaccine doses acquired but not administered was estimated by subtracting the number of doses applied from the total number procured. Based on the analysis, IMSS did not administer approximately 15% of the doses acquired between 2011 and 2023, representing 1,162,582 unutilized doses. By contrast, ISSSTE administered 40% more doses than were officially reported as acquired during the period 2008 to 2023 (329,788 excess doses), and SSA administered 37% more doses than acquired between 2010 and 2023, amounting to 3,453,778 additional doses.

In total, across all institutions and over the full period analyzed (2008–2023), 14% more vaccine doses were administered than confirmed as procured, corresponding to an excess of 2,620,984 doses. These discrepancies may reflect delayed reporting, the use of stock from previous years, or inconsistencies in the procurement records.

## 4. Discussion

This study provides an in-depth evaluation of the HPV vaccination program in Mexico, recognizing inherent limitations due to gaps in data availability across several years.

Despite providing a comprehensive overview of HPV vaccination program performance in Mexico, this study is subject to important limitations. The retrospective and ecological design, combined with the use of secondary data, restricts the depth of analysis and precludes causal inferences. Inconsistencies in data reporting, missing information for several years, and discrepancies between vaccine procurement and administration point to systemic issues in data quality and transparency. These limitations underscore the urgent need for a standardized, centralized, and reliable vaccine information system to improve monitoring, accountability, and strategic planning in national immunization efforts. Variability in data availability across years and institutions—such as missing records from IMSS and ISSSTE for specific periods—further limits the generalizability of the findings. Discrepancies between administered and acquired doses, and coverage estimates exceeding 100% in some years, suggest possible errors in data integration or inventory tracking, underscoring the need for standardized, transparent, and centralized vaccine information systems in Mexico.

The results of this study support the proposed hypothesis, confirming a low level of concordance between official HPV vaccination data and independently derived estimates. The observed discrepancies—such as coverage rates exceeding 100%, years with missing procurement or application data, and inconsistencies between doses acquired and administered—reflect multifactorial weaknesses in the program’s implementation and monitoring. These findings align with the previously documented challenges in Mexico’s vaccination programs for influenza and BCG [7,8], suggesting that systemic issues persist across immunization efforts. Specifically, institutional disparities, inaccurate population estimates, the lack of a nominal registry, and changing vaccination schedules appear to contribute to inconsistent reporting and program performance. Overall, the evidence highlights the need for a coordinated national strategy to address data quality, to improve transparency, and to ensure reliable coverage across all institutions.

Irregularities were identified in annual vaccine acquisition data, characterized by unexplained fluctuations. In theory, these figures should remain relatively stable, given the consistency of the target population year over year. Two exceptions were identified: a notable decline in 2022—coinciding with the transition to a single-dose vaccination strategy—and a significant increase in 2023, when the target population expanded to include three age cohorts. However, the data did not provide sufficient detail to clarify the underlying causes of these variations across all institutions.

An analysis of vaccine administration relative to procurement revealed several years in which the number of doses administered exceeded the number acquired across all institutions. This discrepancy culminated in a total surplus of approximately 2.6 million doses, a figure that cannot be adequately explained with the available data. Potential explanations may include carryover stock from previous years, unrecorded donations, or reporting inconsistencies.

When comparing total vaccines acquired and administered to the theoretical national target population, the results indicated a persistent misalignment. IMSS and ISSSTE consistently reported procurement and coverage levels below target, whereas SSA at times appeared to compensate by acquiring and administering doses in excess of its projected target population. Despite these compensatory efforts, national coverage remained suboptimal in multiple years, with marked year-to-year variability. In total, there were six years in which the necessary vaccines were not procured. Additionally, some data entries are ambiguous, making it unclear whether the reported values represent zero or if the information is entirely missing.

Institution-level comparisons of coverage rates highlighted significant disparities. SSA achieved coverage levels exceeding 100% in eight years, whereas IMSS, and particularly ISSSTE, remained consistently below 70%, with ISSSTE exhibiting the lowest performance overall.

The persistently low coverage rates translated into a considerable number of unvaccinated individuals. Based on target population estimates, approximately 6.8 million doses went unadministered between 2012 and 2023. This shortfall was particularly evident from 2019 to 2021, though a partial recovery was observed in 2022 and 2023.

As illustrated in Figure 7, the calculated coverage rates were compared with those reported by the World Health Organization (WHO) [33]. Although both datasets show a decline in coverage during 2020 and 2021—likely associated with the COVID-19 pandemic—followed by a notable increase in 2022 and 2023, the magnitude of these changes differed significantly between sources, suggesting potential variations in the data collection methods or reporting criteria.

Overall, the findings suggest systemic issues in data reporting, consistency, and completeness. Discrepancies between institutional records, and instances in which more vaccines were administered than officially acquired, point to a lack of standardization across data sources. Furthermore, when comparing the calculated coverage rates with those reported by the World Health Organization (WHO) [33], significant differences were noted. Nevertheless, both sources revealed a consistent pattern: a sharp decline in coverage during 2020 and 2021—likely due to the COVID-19 pandemic—followed by a marked increase in 2022 and 2023.

## 5. Conclusions and Recommendations

This study successfully met its objective of providing a general assessment of the HPV vaccination program in Mexico. The analysis revealed multiple irregularities within the available data, raising concerns about its reliability and reflecting broader issues in the organization and execution of the program. The identified challenges included inconsistencies in defining the target population, discrepancies between vaccine procurement and administration, and potential deficiencies in data quality and reporting practices.

Institutional data showed a lack of alignment between the number of vaccines procured and the estimated size of the theoretical target population, calculated based on each institution’s reported beneficiaries. In several years, the number of administered doses exceeded the number of doses acquired, a discrepancy that was not adequately explained and may indicate reporting errors or inconsistencies in inventory control.

Significant year-to-year variation in the number of vaccine doses purchased was observed, with no clear rationale provided. Furthermore, discrepancies were found between the official coverage reports and the coverage estimates calculated in this study. Adequate vaccination coverage was not consistently achieved, and particularly during the COVID-19 pandemic, a substantial decline in coverage was observed, with millions of doses not administered to the intended population.

A national reorganization of the HPV vaccination program is urgently needed. This requires strong political and operational commitment at all levels. Public health institutions must collaborate to review and update nominal beneficiary data in order to accurately determine the real target population and to ensure effective vaccine delivery. Each institution should take responsibility for identifying its target population, procuring the appropriate number of vaccine doses annually, and implementing an electronic, continuously monitored vaccination registry.

Additionally, a transparent and auditable procurement tracking system should be established. All vaccine-related information—procurement, coverage, shortages, rejections, and distribution—should be publicly available and easily accessible.

To accelerate the goal of cervical cancer elimination, the national strategy should also include catch-up vaccination campaigns for individuals missed in previous years, expansion of vaccine eligibility to additional age cohorts among women, and evaluation of HPV vaccination in males. These efforts must be supported by strengthened programs for early diagnosis and treatment to reduce the burden of HPV-related disease and mortality.

## Figures and Tables

**Figure 1 ijerph-22-01028-f001:**
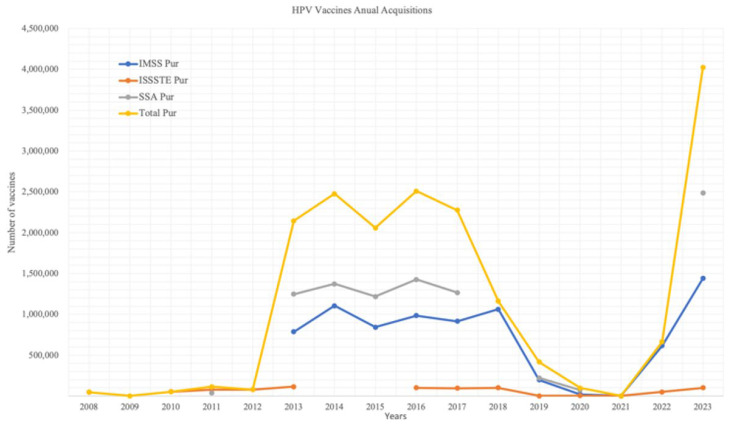
HPV Vaccines Annual acquisitions.

**Figure 2 ijerph-22-01028-f002:**
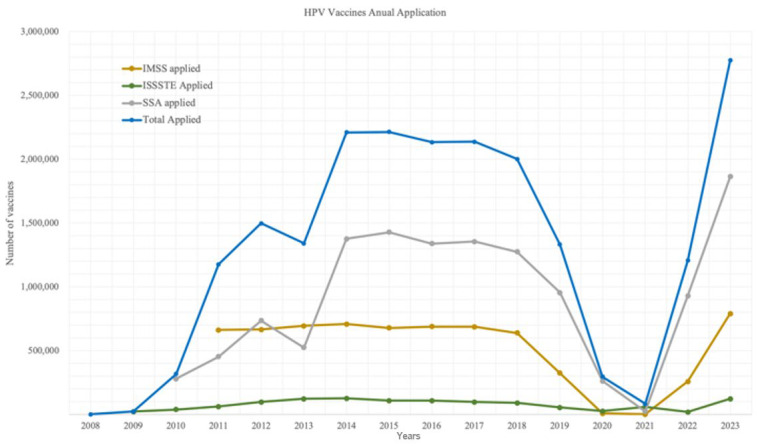
HPV Vaccines Annual Application.

**Figure 3 ijerph-22-01028-f003:**
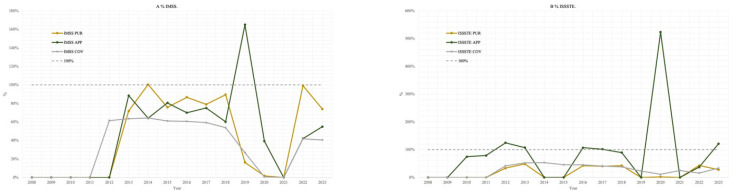
HPV vaccine PUR, APP, and COV: (**A**) % IMSS, (**B**) % ISSSTE, (**C**) % SSA, (**D**) % Total.

**Figure 4 ijerph-22-01028-f004:**
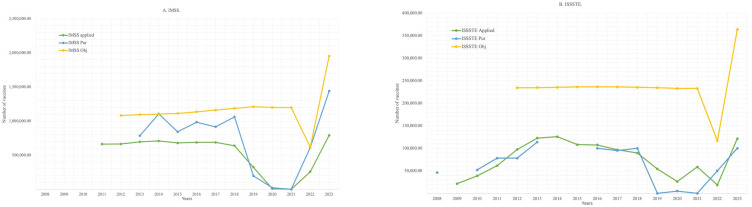
A comparison of HPV vaccine purchases, theoretical target population, and application rates: (**A**) IMSS, (**B**) ISSSTE, (**C**) SSA, (**D**) Total.

**Figure 5 ijerph-22-01028-f005:**
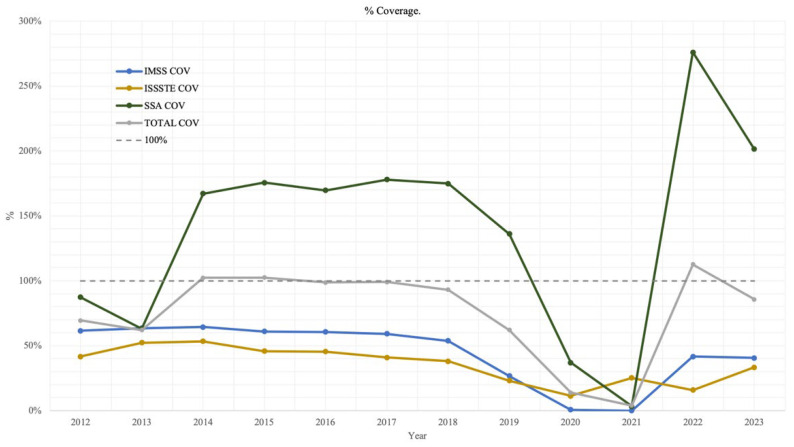
HPV vaccine coverage by institutions and total.

**Figure 6 ijerph-22-01028-f006:**
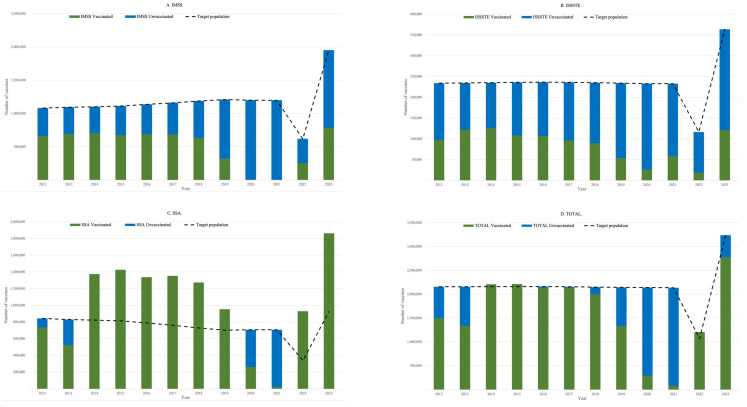
HPV vaccine coverage and proportion of unvaccinated for 2012–2023: (**A**) IMSS, (**B**) ISSSTE, (**C**) SSA, (**D**) Total.

**Figure 7 ijerph-22-01028-f007:**
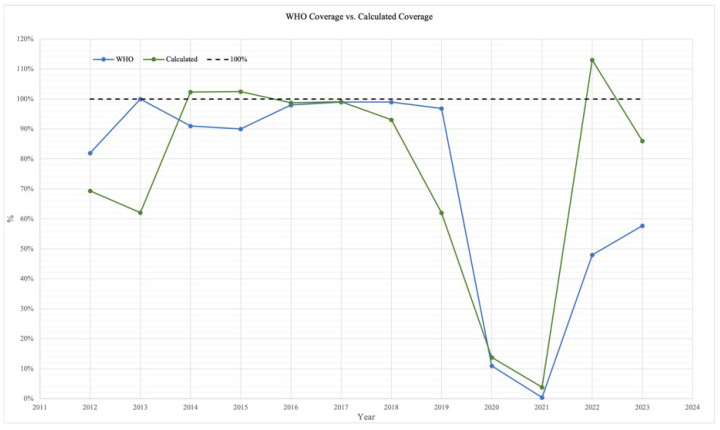
Comparison of WHO Coverage vs. Calculated Coverage.

**Table 1 ijerph-22-01028-t001:** Reported new number of cases in Mexico.

Year	2017	2018	2019	2020	2021	2022	2023
HPVi	22,360	18,316	22,393	6686	10,349	14,118	15,900
CIN1-2	35,321	32,592	34,019	15,929	26,044	32,140	33,705
CIN 3/CIS	4052	3865	4609	2231	3745	5015	5325
CC	2544	3043	3203	2246	2950	3979	4133

HPVi: human papilloma virus infection. CIN1-2: mild and moderate cervical dysplasia. CIN 3/CIS: severe cervical dysplasia and cervical cancer in situ. CC: cervical cancer.

**Table 2 ijerph-22-01028-t002:** Number of women in the population reported (Target population).

Year	2012	2013	2014	2015	2016	2017	2018	2019	2020	2021	2022	2023 *
**Women**	1,078,843	1,079,069	1,079,660	1,080,823	1,079,986	1,076,702	1,073,563	1,071,120	1,068,078	1,087,807	1,071,239	3,305,644

* 10-, 11-, and 12-year-old girls.

**Table 3 ijerph-22-01028-t003:** Percentage of the total population per institution per year (%).

	2008	2009	2010	2011	2012	2013	2014	2015	2016	2017	2018	2019	2020	2021	2022	2023
IMSS	44	44	46	47	49	50	50	51	52	54	55	56	54	56	57	59
ISSSTE	10	10	10	11	11	11	11	11	11	11	11	11	11	11	11	11
Other	2	2	2	2	2	2	1	2	2	1	1	2	2	2	2	2
SSA	44	44	42	40	38	37	38	36	35	34	33	31	33	31	30	28

## Data Availability

The data are available from the sources of each institution cited in the references.

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
