# Peer review of "Sixteen Years of HPV Vaccination in Mexico: Report of the Coverage, Procurement, and Program Performance (2008–2023)"

_ijerph, 2025, doi:10.3390/ijerph22071028_

Round 1
Reviewer 1 Report
Comments and Suggestions for Authors
Sixteen Years of HPV Vaccination in Mexico: Report of the Coverage, Procurement, and Program Performance (2008–2023)
The article analyzes immunization data using data from official Mexican health institutions. It is a topic with relevant practical application as research results can potentially contribute to the improvement of preventive measures to reduce cervical cancer at the national level. The use of institutional data at the national level increases the credibility of the findings. However, several areas require further clarification and elaboration to ensure methodological clarity.
Introduction
- I recommend including the specific objectives of the study, as well as the main hypotheses of the research carried out in a specific sub-item.
- The introduction needs to be shortened.
- Better explain the national HPV vaccination programme.
- What data does each of these institutions collect?
Methods
- Lack of clarity in the methodology, I suggest adding subtitles.
- It is just a suggestion to add the tables as a supplement.
- Please explain how the theoretical target population was calculated, and why you chose the theoretical population instead real population?
- Comparative Analysis Between Institutions: The section implies comparisons across IMSS, ISSSTE, and SSA, but does not specify how institutional differences in population structure, funding, or delivery platforms were accounted for.
- Clarify Yearly Cutoffs: Did the study analyze data by calendar year (January–December), and calculate the numbers of vaccinated and unvaccinated girls in each national cohort?
Results
The results are mostly descriptive, which may be useful, especially for proposing or applying preventive measures at a local level. However, I recommend the authors to perform more comprehensive analyses that add value to the researchHow many doses were acquired annually for each institution and how many doses applied annually?
I suggest adding subtitles: Acquired Doses; Coverage
„The comparison with WHO-Reported Coverage as illustrated in Figure 7, calculated
coverage rates were compared with those reported by the World Health Organization
(WHO) [33]. Although both datasets show a decline in coverage during 2020 and 2021—
likely associated with the COVID-19 pandemic—followed by a notable increase in 2022
and 2023, the magnitude of these changes differed significantly between sources, suggesting
potential variations in data collection methods or reporting criteria“.
Move this part to the discussion, and Figure 7 is unnecessary.
Discussion
The discussion should be further developed by explicitly including whether or not the hypotheses of the study have or have not been met, and by adding references that justify their assertions. The discussion should not be a summary of the results but contrast the data with other studies to identify potential congruencies and discrepancies, and provide an explanation for these.
Add study limitation
I consider that the authors should make some modifications to the manuscript before publication.

Author Response
Comments 1: Introduction • I recommend including the specific objectives of the study, as well as the main hypotheses of the research carried out in a specific sub-item. • The introduction needs to be shortened. • Better explain the national HPV vaccination programme. • What data does each of these institutions collect?
Response 1: The introduction was cut and the changes were made to the manuscript and are marked in yellow.
Comments 2: Methods • Lack of clarity in the methodology, I suggest adding subtitles. • It is just a suggestion to add the tables as a supplement. • Please explain how the theoretical target population was calculated, and why you chose the theoretical population instead real population? • Comparative Analysis Between Institutions: The section implies comparisons across IMSS, ISSSTE, and SSA, but does not specify how institutional differences in population structure, funding, or delivery platforms were accounted for. • Clarify Yearly Cutoffs: Did the study analyze data by calendar year (January–December), and calculate the numbers of vaccinated and unvaccinated girls in each national cohort?
Response 2: the changes were made to the manuscript and are marked in yellow. A paragraph was added on limitations an hypothesis in the methodology and discussion.
Comments 3: Results The results are mostly descriptive, which may be useful, especially for proposing or applying preventive measures at a local level. However, I recommend the authors to perform more comprehensive analyses that add value to the researchHow many doses were acquired annually for each institution and how many doses applied annually?
Response 3: Some changes were made to clarify the results, are marked in yellow.
Comments 4:
Move this part to the discussion, and Figure 7 is unnecessary.
Discussion
The discussion should be further developed by explicitly including whether or not the hypotheses of
the study have or have not been met, and by adding references that justify their assertions. The
discussion should not be a summary of the results but contrast the data with other studies to identify
potential congruencies and discrepancies, and provide an explanation for these.
Add study limitation
I consider that the authors should make some modifications to the manuscript before publication.
Response 4: Added, marked in yellow, in the discussion whether or not the hypotheses were fulfilled, the comparative figure and other comparisons with the literature
Reviewer 2 Report
Comments and Suggestions for Authors
Feregrino et al., aimed to evaluate the current status of the vaccination program in Mexico, analyzing trends in vaccine acquisition, administration, and coverage over a 16-year period. The topic is fascinating. However, I have provided my suggestions here to improve the quality of the work:
Introduction:
1) “The first prophylactic vaccine targeting Human Papillomavirus (HPV)-associated diseases was authorized for use in 2006”: please specify that the authorization for the use of HPV vaccine was granted in 2006 in the USA and Europe [Quadrivalent HPV L1 VLP vaccine (Gardasil-4)].
2) “All licensed HPV vaccines are approved for use in both females and males aged 9 years and older”: This is partially correct because there is significant variation in the design of national vaccination programmes (W. Wang et al. 2022, doi: 10.1080/14760584.2022.2129615). In Italy, for example, HPV vaccination is authorized starting from 12 years of age, according to the following regimen: two doses up to 15 years old; three-dose schedule (at 0, 1–2, and 6 months) at 15–45 years old. Thus, please specify where HPV vaccination is authorized starting from 9 years of age.
3) “Current scientific evidence supports the adoption of a two-dose schedule for HPV vaccination in the primary target population beginning at 9 years of age (…)”: Are the authors referring to the vaccination schedule in Mexico? Please specify, as this is not clear, and include an appropriate reference.
4) The Table1 is not interpretable in its current form. Please specify the units of measurement.
Material and Methods:
1) “The study period covered 15 years, from 2008 through 2023”: please specify if the study covered a period of 15 or 16 years.
2) Specify the difference between “Number of vaccine doses acquired” and “Number of doses administered”.
3) Table2 and 3 are not interpretable in the current forms. Please specify the units of measurement and include the definitions of the acronyms in the footnotes.
4) “Regarding HPV vaccination, official recommendations in Mexico indicated vaccination for 11-year-old girls through 2022 [2]. In 2023, these indications were expanded to include girls aged 10, 11, and 12 years”: please add this information in Introduction where appropriate.
Results:
Please improve the resolution of figures to ensure clarity and readability.
Discussion:
It would be reasonable to include a section at the end of the discussion outlining the strengths and limitations of the study.
Author Response
Introduction:
- “The first prophylactic vaccine targeting Human Papillomavirus (HPV)-associated diseases was authorized for use in 2006”: please specify that the authorization for the use of HPV vaccine was granted in 2006 in the USA and Europe [Quadrivalent HPV L1 VLP vaccine (Gardasil-4)].
Added to the manuscript in yellow
2) “All licensed HPV vaccines are approved for use in both females and males aged 9 years and older”: This is partially correct because there is significant variation in the design of national vaccination programmes (W. Wang et al. 2022, doi: 10.1080/14760584.2022.2129615). In Italy, for example, HPV vaccination is authorized starting from 12 years of age, according to the following regimen: two doses up to 15 years old; three-dose schedule (at 0, 1–2, and 6 months) at 15–45 years old. Thus, please specify where HPV vaccination is authorized starting from 9 years of age.
The text was removed to shorten the introduction.
3) “Current scientific evidence supports the adoption of a two-dose schedule for HPV vaccination in the primary target population beginning at 9 years of age (…)”: Are the authors referring to the vaccination schedule in Mexico? Please specify, as this is not clear, and include an appropriate reference.
The text was removed to shorten the introduction.
4) The Table1 is not interpretable in its current form. Please specify the units of measurement.
Added to the tablle 1 title “Reported new number of cases in Mexico”.
Material and Methods:
1) “The study period covered 15 years, from 2008 through 2023”: please specify if the study covered a period of 15 or 16 years.
Changed to 16 years
2) Specify the difference between “Number of vaccine doses acquired” and “Number of doses administered”.
2 words were added to be clearer
o Number of vaccine doses acquired or purchased,
o Number of doses administered or apply,
3) Table2 and 3 are not interpretable in the current forms. Please specify the units of measurement and include the definitions of the acronyms in the footnotes.
Table 2 Added: (Target population).
Table 3 Added: IMSS: The Mexican Social Security Institute, ISSSTE: The Institute for Social Security and Services for State Workers, SSA: The Ministry of Health. * Each number represents the percentage of population by institution.
4) “Regarding HPV vaccination, official recommendations in Mexico indicated vaccination for 11-year-old girls through 2022 [2]. In 2023, these indications were expanded to include girls aged 10, 11, and 12 years”: please add this information in Introduction where appropriate.
changed to introduction
Results:
Please improve the resolution of figures to ensure clarity and readability.
Images are sent separately for better resolution.
Discussion:
It would be reasonable to include a section at the end of the discussion outlining the strengths and limitations of the study.
A yellow paragraph was added to the methodology and discussion sections.
